# Damage Analysis of CFRP Hybrid Bonded-Bolted Joint during Insertion of Interference-Fit Bolt

**DOI:** 10.3390/ma16103753

**Published:** 2023-05-15

**Authors:** Long Yan, Ruisong Jiang, Yangjie Zuo

**Affiliations:** 1School of Mechanical and Electrical Engineering, Xi’an University of Architecture and Technology, Xi’an 710055, China; huanglong731@xauat.edu.cn; 2School of Mechanical Engineering, Sichuan University, Chengdu 610065, China; 3School of Aeronautics and Astronautics, Sichuan University, Chengdu 610065, China; zuoyangjie@scu.edu.cn

**Keywords:** CFRP, HBB joint, interference-fit size, damage, insertion force

## Abstract

In this study, experiments and finite element analysis (FEA) were used to evaluate the impact of interference-fit sizes on CFRP hybrid bonded-bolted (HBB) joint damage during bolt insertion. The specimens were designed in accordance with the ASTM D5961 standard and bolt insertion tests were performed at selected interference-fit sizes (0.4%, 0.6%, 0.8%, and 1%). Damage to composite laminates was predicted using the Shokrieh–Hashin criterion and Tan’s degradation rule via the user subroutine USDFLD, while damage to the adhesive layer was simulated by the Cohesive Zone Model (CZM). The corresponding bolt insertion tests were performed. The variation of insertion force with interference-fit size was discussed. The results showed that matrix compressive failure was the main failure mode. With the growth of the interference-fit size, more failure modes appeared, and the failure region expanded. Regarding the adhesive layer, it did not completely fail at the four interference-fit sizes. This paper will be helpful in designing composite joint structures and especially for understanding CFRP HBB joint damage and failure mechanisms.

## 1. Introduction

Due to their exceptional properties (lightweight, high-strength, flexible, and anti-corrosive), Carbon Fiber Reinforced Polymer (CFRP) composites are becoming increasingly attractive as the next generation of structural materials for aerospace applications [1,2]. Although composite structures have evolved toward integrated design and manufacturing, effective joining techniques for composites remain inevitable because of complex structural and dimensional constraints [3,4]. Mechanically fastened joints and adhesive-bonded joints are the most common joining methods used in composite structures. Given the characteristics of these two methods, researchers have wanted to combine them to achieve better performance. In Hart-Smith’s study [5], a combined (bolted and bonded) joint was found to be no stronger than a fully bonded joint in terms of strength, but it was useful in repairing damaged adhesives and preventing damage extension. For aerospace-grade joints using high-modulus adhesives and large overlap areas, the load-bearing capacity of the adhesive has been seen to be sufficient and hybrid joints are considered unnecessary [6]. On the contrary, the careful design of combination hybrid joints with low-modulus adhesives allows both the bolt and the adhesive to carry the load, resulting in stronger joints and better fatigue life [7]. The hybrid bonded-bolted (HBB) joint with interference fit is often obtained by pressing a larger bolt into a smaller joint hole in the bonded specimen [8,9]. However, the insertion of interference-fit bolts may cause varying degrees of damage in the bonded and bolted areas. This damage can lead to localized degradation of the material properties, which in turn affects the mechanical properties of the entire structure [10,11]. Thus, a study of how CFRP HBB joints are damaged and fail during bolt insertion is essential.

To the authors’ best knowledge, most of the literature is devoted to the study of the performance of purely bolted, purely bonded, and hybrid joint structures, and, in particular, to the influence of structural parameters on the mechanical properties of HBB joints [12,13,14,15]. Despite the advantages of the interference fit in improving static and fatigue strength, improper interference-fit sizes can lead to joint damage or even delamination of joint laminates [16,17,18,19], especially when inserting interference-fit bolts. Only a few works of literature reported the damage and failure behaviors during interference-fit bolt insertion. Through experiments and simulations, Li et al. [20] investigated how the interference-fit size influenced the damage of bolt jointed single lap CFRP/Titanium composite structures. The results showed that proper interference-fit sizes could make CFRP holes “brush-like”, exhibiting “softening” and “cushioning”, thus increasing load-bearing capacity. Jiang et al. [21] carried out insertion tests on aluminum alloy specimens using titanium alloy high-locking bolts and developed a 2D finite element model for static simulation. The study systematically and quantitatively analyzed the insertion force variation, hole deformation, and residual stress distribution during bolt insertion. Zuo et al. [22] experimentally investigated the damage to CFRP joints with interference-fit sizes in the process of inserting bolts. It was found that the upper plate was most damaged at the inlet and around the hole and that the damage increased with increasing interference-fit size. In addition, fiber and matrix breaks were observed at the outlet of the hole in the lower plate. Hu et al. [9,23] used interference-fit bolt insertion tests to study the interfacial properties of composite joints during assembly. The findings indicated that the insertion of bolts with proper interference-fit sizes provided cold expansion that contributed to the formation of a strong bonding interface and maintained a constant and uniform residual stress around the joint hole. Zhang et al. [24] pointed out that bolt loads had a greater effect on the load-carrying capacity of single-lap interference-fit composite joints than interference-fit sizes, and that excessive bolt loads and interference-fit sizes could lead to damage to the joint-hole surface. Kouka et al. [25] studied the failure behavior of Carbon/PPS composite plates with different configurations of interaction holes under tensile loading. They found that the location of the hole and the orientation of the laminate were the main factors affecting the stress concentration around the hole. Nassiraei et al. [26] investigated the effect of material parameters and joint shape on the stress concentration factor (SCF) of a chord member on a tubular X-shaped joint reinforced with fiber-reinforced polymer (FRP) by means of a finite element method. They also studied the initial stiffness, ultimate capacity, capacity ratio, and failure mechanism of tubular X-shaped joints reinforced with FRP under compressive loading [27]. Some scholars have conducted a series of studies on simulation strategies for bolted joints. Belardi et al. [28] developed a Composite Bolted Joint Element (CBJE) through a user-defined finite element method and applied it to the analysis of single-lap, single-bolt composite bolted joints. Sharos et al. [29] used a user-defined finite element method to simplify bolted joints and completed simulations of composite bolted joints at different loading rates. Liu et al. [30] proposed an improved 2D finite element model taking into account secondary bending to predict the load distribution in single-lap, multi-bolt composite joints. In comparison with 3D modeling methods, it was demonstrated that these methods greatly reduce the computational effort while ensuring the accuracy of the results.

In this study, the damage of CFRP HBB joints during interference-fit bolt insertion was investigated numerically and experimentally. Specifically, the user subroutine USDFLD was used to predict the failure modes and regions of CFRP plates, and the CZM was used to simulate the adhesive layer between the upper and lower plates. Accordingly, bolt insertion tests with different interference-fit sizes were performed to verify the numerical results. The findings of this research will benefit the development and utilization of CFRP HBB joints.

## 2. Numerical Simulation

### 2.1. Problem Statements

The specimen design for the interference-fit bolt insertion test was based on the ASTM D5961 standard, as shown in Figure 1. The length of the CFRP plate was 90 mm, the width was 30 mm, and the hole edge distance was 15 mm. It was made of T800/epoxy with a stacking sequence of [45/−45/0/45/90/−45/45/90/−45]_s_, and its mechanical properties are shown in Table 1. Since the nominal thickness of each layer was 0.188 mm, the nominal thickness of the laminate was 3.384 mm. Considering the accuracy and efficiency of the simulation, it was necessary to simplify the bolt and ignore the threaded part. The bolt shank had a diameter of 5 mm and the bolt length was set to 15 mm. The bolt was made of titanium alloy. It had an elastic modulus of 110 GPa and a Poisson’s ratio of 0.34. This bolt was then pressed into the joint hole using a steel indenter (elastic modulus 210 GPa, Poisson’s ratio 0.3). To be consistent with the test conditions, a steel block was placed under the specimen which had a through hole of 5.2 mm diameter in the center. The presence of this hole ensured that the bolt would not interfere with the block after being pressed out. The upper and lower CFRP plates were bonded by a bi-component adhesive, Hysol EA9361. The adhesive layer had a thickness of 0.1 mm and its performance parameters are summarized in Table 2.

### 2.2. Finite Element Modeling in ABAQUS/Standard

A 3D finite element model of the specimen was developed in ABAQUS/Standard, as illustrated in Figure 2. The 8-node linear brick, reduced integration, and hourglass control (C3D8R) elements were applied to the CFRP plates as well as to the metal parts of the model. The modeling of one layer element representing one ply provided an accurate description of the CFRP plate. Thus, there were a total of 18 plies in the CFRP plate. The adhesive layer was modeled by the 8-node 3D cohesive (COH3D8) elements, divided into only one layer in the thickness direction. This model was used to analyze the insertion forces of the bolts and to predict the damage and failure of the CFRP HBB joints. Therefore, a mesh refinement of the area around the hole was required. The CFRP plate was divided into two regions: an overlap region and a non-overlap region. The overlap region used a fine mesh with in-plane dimensions of 0.2 mm × 0.4 mm to 1 mm × 1 mm elements, while the non-overlap region used a relatively coarse mesh (2 mm × 1 mm). This meshing method did not cause convergence problems and had high accuracy while ensuring computational efficiency. The meshing of the adhesive layer was the same as the meshing of the overlap region of the CFRP plate. A total of 134,736 elements were included in the model of the CFRP HBB joint, of which 96,480 elements were used to model the upper and lower CFRP plates, 7120 elements were used to model the adhesive layer, and 31,136 elements were used for the other metal parts. The bolt, indenter, and block were modeled as rigid bodies, while the CFRP plate and the adhesive layer were deformable. In keeping with the test, the bolt was inserted a short distance into the hole in the upper plate. This modeling approach achieved an initial interference fit between the bolt and the hole.

### 2.3. Contact Relationships and Boundary Conditions

As shown in Figure 3, two tie relationships and four contact pairs were established in the model. The two tie relationships were the upper plate and adhesive layer (tie 1) and the lower plate and adhesive layer (tie 2). The four contact pairs were the indenter and bolt (contact 1), bolt and hole wall (contact 2), bolt and adhesive layer (contact 3), and block and lower plate (contact 4). A surface-to-surface discretization method and the corresponding contact properties were then applied to these contacts. Considering the large relative sliding between the bolt and the hole during the insertion of the bolt, a finite sliding formulation was used in contact 2 and contact 3, while a small sliding formulation was used in contact 1 and contact 4. The normal behavior used a “hard” contact, while the tangential behavior used a penalty function. The friction coefficients of contact 2 and contact 3 were set to 0.1 as suggested by Kim [33] and McCarthy [34]. For contact 1 and contact 4, the friction coefficient was set to 0.3 to eliminate rigid body displacement.

Considering the test setup, fixed constraints were applied to the left surface of the lower plate, the right surface of the upper plate, and the bottom surface of the block. In addition, the bolt was inserted into the hole of the CFRP plate under the pressure of the indenter. Thus, four boundary conditions were established in the model, including three fixed constraints and the indenter displacement along the z-axis. As mentioned before, the bolt length was set to 15 mm, while the total thickness of the two CFRP plates and the adhesive layer was 6.868 mm. Therefore, the indenter displacement was set to 10 mm to ensure that the bolt could completely penetrate the lower plate. Two steps (Step 1 and Step 2) were created in this model with the aim of establishing a stable contact and improving convergence. Specifically, a tiny displacement of 0.001 mm along the z-axis was applied to the indenter in Step 1 and the full displacement load (10 mm) was added in Step 2.

### 2.4. Implementation of Interference Fit in ABAQUS/Standard

In ABAQUS/Standard, the interference fit could be achieved in two ways: geometric modeling or contact setting. For the first method, the bolt and hole were modeled based on their actual diameters. That was to say, the interference fit was achieved by pressing a larger-diameter bolt into a smaller-diameter hole. For the second method, the bolt and hole were modeled using the bolt diameter, and then the interference fit was obtained by setting the “interference fit” option in the interaction module. Considering the easy modification of sizes, the second method was used in this study to achieve different interference-fit sizes. According to the ABAQUS documentation on “interference fit” content, an amplitude reference should be specified to define a particular allowable interference-time variation. By default, the value of the specified allowable interference was applied instantaneously at the start of the step and then ramped down to zero linearly over the step. During the insertion of an interference-fit bolt, the bolt applied the full interference value at the beginning and kept the interference value it provided constant. Obviously, the default amplitude reference did not meet the requirements. Therefore, it was necessary to redefine a reference whose amplitude was always 1.

### 2.5. Failure Prediction Formulations for Composite Laminates

In this study, the Shokrieh–Hashin criterion, based on the 3D Hashin criterion, was used to predict the laminate damage in CFRP HBB joints. Seven failure modes were included in this failure criterion: fiber tensile/compressive failure, fiber–matrix shear damage, matrix tensile/compressive failure, and delamination in tension/compression. These failure modes are defined separately as [35]:

Fiber tensile failure (*σ*_11_ ≥ 0):(1)$$e_{\mathrm{FT}}^{2}={\left(\frac{\sigma_{11}}{X_{T}}\right)}^{2}+{\left(\frac{\tau_{12}}{S_{12}}\right)}^{2}+{\left(\frac{\tau_{13}}{S_{13}}\right)}^{2}=\left\{\begin{array}{l} \begin{array}{cc} \geq 1 & \mathrm{Failure} \end{array}\\ \begin{array}{cc} <1 & \mathrm{No}\;\mathrm{failure} \end{array} \end{array}\right.$$

Fiber compressive failure (*σ*_11_ < 0):(2)$$e_{\mathrm{FC}}^{2}={\left(\frac{\sigma_{11}}{X_{C}}\right)}^{2}=\left\{\begin{array}{l} \begin{array}{cc} \geq 1 & \mathrm{Failure} \end{array}\\ \begin{array}{cc} <1 & \mathrm{No}\;\mathrm{failure} \end{array} \end{array}\right.$$

Fiber–matrix shear damage (*σ*_11_ < 0):(3)$$e_{\mathrm{FMS}}^{2}={\left(\frac{\sigma_{11}}{X_{C}}\right)}^{2}+{\left(\frac{\tau_{12}}{S_{12}}\right)}^{2}+{\left(\frac{\tau_{13}}{S_{13}}\right)}^{2}=\left\{\begin{array}{l} \begin{array}{cc} \geq 1 & \mathrm{Failure} \end{array}\\ \begin{array}{cc} <1 & \mathrm{No}\;\mathrm{failure} \end{array} \end{array}\right.$$

Matrix tensile failure (*σ*_22_ ≥ 0):(4)$$e_{\mathrm{MT}}^{2}={\left(\frac{\sigma_{22}}{Y_{T}}\right)}^{2}+{\left(\frac{\tau_{12}}{S_{12}}\right)}^{2}+{\left(\frac{\tau_{23}}{S_{23}}\right)}^{2}=\left\{\begin{array}{l} \begin{array}{cc} \geq 1 & \mathrm{Failure} \end{array}\\ \begin{array}{cc} <1 & \mathrm{No}\;\mathrm{failure} \end{array} \end{array}\right.$$

Matrix compressive failure (*σ*_22_ < 0):(5)$$e_{\mathrm{MC}}^{2}={\left(\frac{\sigma_{22}}{Y_{C}}\right)}^{2}+{\left(\frac{\tau_{12}}{S_{12}}\right)}^{2}+{\left(\frac{\tau_{23}}{S_{23}}\right)}^{2}=\left\{\begin{array}{l} \begin{array}{cc} \geq 1 & \mathrm{Failure} \end{array}\\ \begin{array}{cc} <1 & \mathrm{No}\;\mathrm{failure} \end{array} \end{array}\right.$$

Delamination in tension (*σ*_33_ ≥ 0):(6)$$e_{\mathrm{DT}}^{2}={\left(\frac{\sigma_{33}}{Z_{T}}\right)}^{2}+{\left(\frac{\tau_{13}}{S_{13}}\right)}^{2}+{\left(\frac{\tau_{23}}{S_{23}}\right)}^{2}=\left\{\begin{array}{l} \begin{array}{cc} \geq 1 & \mathrm{Failure} \end{array}\\ \begin{array}{cc} <1 & \mathrm{No}\;\mathrm{failure} \end{array} \end{array}\right.$$

Delamination in compression (*σ*_33_ < 0):(7)$$e_{\mathrm{DC}}^{2}={\left(\frac{\sigma_{33}}{Z_{C}}\right)}^{2}+{\left(\frac{\tau_{13}}{S_{13}}\right)}^{2}+{\left(\frac{\tau_{23}}{S_{23}}\right)}^{2}=\left\{\begin{array}{l} \begin{array}{cc} \geq 1 & \mathrm{Failure} \end{array}\\ \begin{array}{cc} <1 & \mathrm{No}\;\mathrm{failure} \end{array} \end{array}\right.$$
where *σ*_ii_ (ii = 11, 22, 33) is the normal stress; *τ*_ij_ (ij = 12, 13, 23) is the shear stress; *X*_T_, *Y*_T_, and *Z*_T_ are the tensile strengths in the longitudinal, transverse, and thickness directions; *X*_C_, *Y*_C_, and *Z*_C_ are the compressive strengths in the longitudinal, transverse and thickness directions; and *S*_ij_ (ij = 12, 13, 23) is the shear strength.

By using failure criteria, only the damage initiation of composite laminates was addressed. Once damage occurred, degradation of the material performance parameters was required. In progressive failure analysis, material property degradation models can be divided into three categories: transient degradation, progressive degradation, and failure element constant stress state. Transient degradation instantaneously reduces the relevant material parameters to zero or a fraction of their original values, progressive degradation gradually (linearly or exponentially) reduces the material properties to zero, and the failure element’s constant stress state means that the element cannot be subjected to additional loads. The first degradation model was easy to implement and accurately predicted the material properties after damage evolution. Therefore, a transient degradation model with reference to Tan’s degradation rules was used in this study [36] (see Table 3).

Failure analysis and material property degradation were implemented by using the user subroutine USDFLD. In the subroutine, property degradation was controlled by seven field variables (FVs). FV1, FV2: fiber tensile/compressive failure; FV3: fiber–matrix shear damage; FV4, FV5: matrix tensile/compressive failure; and FV6, FV7: delamination in tension/compression. Initially, the field variables at all integration points were given values of zero and the material parameters were set to their original values. Then, external loads were gradually applied, and calculations were performed in each increment using an iterative method until the equilibrium state of the analysis converged. After each increment, the failure flags (*e*_FT_, *e*_FC_, *e*_FMS_, *e*_MT_, *e*_MC_, *e*_DT_, and *e*_DC_) corresponding to Equations (1)–(7) were calculated at each element’s integration points. The results of failure flags were not assigned directly to FVs but were stored as solution-dependent state variables (SDVs). Once an SDV exceeded 1, the associated FV was assigned a value of 1 and remained constant throughout the simulation. When the value of FV was equal to 1, the corresponding property degradation was performed by the rules in Table 3. The analysis process was terminated when excessive element distortion occurred or when the design load was reached.

### 2.6. Adhesive Layer Damage Theory

To simulate the interfacial debonding of CFRP HBB joints, the CZM defined by the traction–separation law was used in this study. The bilinear constitutive model was a typical traction–separation response, as shown in Figure 4. *T*_0_ denoted the original thickness of the cohesive element, and *δ*_n_, *δ*_s_, and *δ*_t_ denoted the components of the separation displacement; the nominal strains were then defined as:(8)$$\left\{\begin{array}{l} \varepsilon_{n}=\delta_{n}/T_{0}\\ \varepsilon_{s}=\delta_{s}/T_{0}\\ \varepsilon_{t}=\delta_{t}/T_{0} \end{array}\right.$$

The degradation process started when the specified damage initiation criteria were reached. There were several damage initiation criteria available in ABAQUS. In this study, a quadratic interaction function involving the nominal stress ratios was used for the initial damage of the adhesive layer. This criterion was expressed as:(9)$${\left(\frac{\langle t_{n}\rangle }{t_{n}^{0}}\right)}^{2}+{\left(\frac{t_{s}}{t_{s}^{0}}\right)}^{2}+{\left(\frac{t_{t}}{t_{t}^{0}}\right)}^{2}=1$$
where ⟨⟩ is the Macaulay bracket, signifying that a pure compressive deformation or stress state does not initiate damage; *t*_n_, *t*_s_, and *t*_t_ are the nominal stress components; and *t_n_*^0^, *t_s_*^0^, and *t_t_*^0^ represent the peak of nominal stresses for deformation in the pure normal, pure first, and second shear directions, respectively.

After the damage had occurred, the elastic behavior could be expressed as:(10)$$\left\{\begin{array}{c} t_{n}\\ t_{s}\\ t_{t} \end{array}\right\}=\left(1-d\right)\left[\begin{array}{ccc} E_{\mathrm{nn}} & & \\ & E_{\mathrm{ss}} & \\ & & E_{\mathrm{tt}} \end{array}\right]\left[\begin{array}{c} \varepsilon_{n}\\ \varepsilon_{s}\\ \varepsilon_{t} \end{array}\right]$$
where *d* is a scalar damage variable. It has an initial value of 0 and increases from 0 to 1 upon further loading after the damage has occurred. By selecting SDEG in the ABAQUS field output, the stiffness degradation of the adhesive layers can be presented in post-processing.

The damage evolution could be defined according to the fracture energy, and the displacement at complete failure could be determined by the B–K criterion.
(11)$$G^{C}=G_{n}^{C}+\left(G_{s}^{C}-G_{n}^{C}\right){\left(\frac{G_{s}+G_{t}}{G_{n}+G_{s}+G_{t}}\right)}^{\eta }$$
where *G*^C^, *G*_n_^C^, and *G*_s_^C^ are the total, normal, and shear critical fracture energies, respectively; *G*_n_, *G*_s_, and *G*_t_ are the work completed by the tractions in the normal, first, and second shear directions, respectively; and *η* is a material parameter (*η* = 2).

## 3. Experimental Details

### 3.1. Test Piece Preparation

Table 4 lists the geometric and material data of the insert specimens used for the tests. The mechanical properties of T800/epoxy and Hysol EA9361 are shown in Table 1 and Table 2, respectively.

The interference–fit relationship was established in the specimen by inserting a standard-size bolt into an undersized joint hole. Thus, the definition of the interference-fit size could be expressed as:(12)$$I=\frac{D_{1}-D_{0}}{D_{0}}\times 100\%$$
where *D*_1_ and *D*_0_ are the diameters of the bolt and the hole, respectively.

As shown in Table 5, 0.4%, 0.6%, 0.8%, and 1% interference-fit sizes were selected for the bolt insertion tests. Substituting these interference-fit sizes and the standard bolt diameter of 5 mm into Equation (12), the corresponding nominal hole diameters were calculated to be 4.980 mm, 4.970 mm, 4.960 mm, and 4.950 mm, respectively. Considering the difference between the nominal and real values of the diameter, the bolt and hole need to be measured and matched to obtain the actual interference-fit sizes required. In this study, the matching method in [9] was consulted. For each size of an interference fit, a lower-than-nominal-size bolt was selected to correspond to a lower-than-nominal-size hole within the tolerance range, and the same is true for higher-than-nominal-size bolts and holes. Thus, four interference-fit intervals were obtained, which were 0.25%–0.57%, 0.34%–0.68%, 0.56%–0.87%, and 0.77%–1.06%.

### 3.2. Test Procedure

The bolt insertion test was conducted on the ETM101B material testing machine (Sichuan Dexiang Kechuang Instrument Co., Ltd., Chengdu, China). The test setup and insertion specimen are shown in Figure 5. The threaded part of the bolt could be manually screwed into the joint hole in advance, as its diameter was smaller than the bolt shank’s diameter. This created a guide when inserting the bolt, ensuring verticality and coaxiality. Furthermore, the uniform distribution of forces around the joint hole also minimized damage to the CFRP plate [23]. The bolt was pressed into the joint hole by the testing machine at a rate of 3 mm/min until the bolt shank completely penetrated the lower plate. The load–displacement data were recorded in real-time. The length of the bolt shank was around 7 mm, which was greater than the total thickness of the two CFRP plates and the adhesive layer (6.868 mm). This allowed the bolt shank and the joint hole to maintain an interference fit throughout the bolt insertion process.

## 4. Results and Discussion

### 4.1. Comparison of Insertion Force Results

During the process of inserting an interference-fit bolt, the press-in force consists of two parts: the hole deformation force caused by the bolt insertion and the friction force between the bolt and the hole. This force determines to some extent the deformation of the hole and the stress distribution around the hole. Figure 6 shows the experimental and simulated load–displacement curves at 0.4%, 0.6%, 0.8%, and 1% interference-fit sizes.

Affected by dimensional errors, material defects, and some random factors in the test, the four groups of experimental curves exhibited the same characteristics except for differences in magnitudes. As the insertion depth gradually increased, the insertion force also steadily increased. During this process, the bolt must enlarge the joint hole and overcome the resulting increasing interfacial friction. Once the bolt shank arrived at the adhesive layer between the two plates, the insertion force exhibited a slight decrease due to the small deformation that occurred in the lower plate. The bolt was then inserted further into the lower plate and the insertion force continued to increase. After the bolt completely penetrated the lower plate, the resistance came only from the interfacial friction. As a result, a second decline in force occurred. For the 0.4%, 0.6%, 0.8%, and 1% interference-fit tests, the average values of the maximum insertion forces were 1348 N, 2112 N, 2797 N, and 3461 N, respectively.

The experimental and numerical results of the peak insertion force were then compared, as shown in Table 6. Some relative errors existed in the results, with a maximum error of 8.73%. In the FEA, the bolt shank was pressed completely vertically into the joint hole while idealized assumptions were made about the interfacial friction. Actually, the friction force was not exactly the same throughout the hole wall, and there was some error in the shape and position of the joint hole. In addition, small deflections may occur during bolt insertion. The test results of the insertion force basically agree with the FEA results, which also verified the correctness of the established numerical model.

### 4.2. Damage Analysis around the Joint Hole

This section focuses on the damage and failure mechanisms around the hole, including the sequence in which composite failure occurred, the expansion of composite failure regions, and the influence of interference-fit sizes on failure modes and adhesive layer damage.

The failure sequence of the composite laminate around the hole at 1% interference-fit size was analyzed, as shown in Figure 7. As mentioned before, seven failure modes were considered in this study. For ease of expression, fiber tensile/compressive failure is abbreviated as FT/FC, fiber–matrix shear damage is abbreviated as FMS, matrix tensile/compressive failure is abbreviated as MT/MC, and delamination in tension/compression is abbreviated as DT/DC. MT was observed as the first failure mode of the upper plate. The failure occurred at 0 s and was due to the fact that the bolt and hole had already made initial interference-fit contact. After pressing the bolt into the upper plate for a short distance, the hole wall was squeezed and deformed. Thus, at 0.02 s, FMS and MC occurred almost simultaneously in the upper plate. As the bolt was gradually inserted, the joint hole continued to expand. This exacerbated the damage to the composite laminate around the hole and caused an increasing number of failure modes to occur. At 0.04 s and 0.12 s, DT and DC were observed successively in the upper plate. Similar deformation occurred at the entrance of the lower plate when the bolt shank passed through the interfacial layer between the two plates, resulting in MC and FMS in the lower plate at 0.38 s and 0.4 s, respectively. It was noteworthy that no fiber failure was found in either the upper or lower CFRP plates at the interference-fit size of 1%.

The CFRP failure region of the upper plate continued to expand with the increase in bolt insertion depth, as shown in Figure 8. The MC of the upper plate started at the topmost surface (45° ply). At an insertion depth of 0.6 mm, a small number of failure elements were observed in the 0° and −45° plies. The failure region of the 0° and −45° plies expanded rapidly with increasing bolt insertion depth. When the insertion depth reached 1.6 mm, the failure region spread down the axial direction of the joint hole to the 45° and −45° plies. Thereafter, although the insertion depth was still increasing, the number of added failure elements was significantly reduced. After the bolt fully penetrated the upper plate, only very few added failure elements were found in the vicinity of the upper plate exit. It could be seen that during bolt insertion, the MC expanded along the axial and circumferential directions of the joint hole. This was due to the fact that, in addition to the damage from radial compression, the interfacial friction also caused axial compression of the laminate. The continuous press-in of the bolt not only increased the contact region between the bolt and the hole but also strengthened the axial pressure on the laminate, which eventually led to the axial and circumferential expansion of the failure region.

Figure 9 presents the 3D distribution of the failure regions for different failure modes at interference fit sizes of 0.4%, 0.6%, 0.8%, and 1%. None of the failure modes occurred at an interference-fit size of 0.4% due to the slight deformation of the joint hole and the small bolt-hole interfacial friction. At the 0.6% interference-fit size, MT occurred near the upper plate entrance. For the 0.8% interference-fit size, in addition to MT, FMS and MC were also observed in the upper plate. In the case of the 1% interference-fit size, the number of failure modes was five; the two extra failure modes were DT and DC. Meanwhile, FMS and MC also occurred in the lower plate. However, no fiber failure was observed at the interference-fit sizes selected in this study due to the fact that the fiber strength was much higher than the matrix strength. The failure of the CFRP during bolt insertion was the result of bolt pressure and friction between the bolt and the hole. The growth in interference-fit size caused an increase in the number of failure modes and resulted in a gradual expansion of the failure region from the upper plate to the lower plate. These failures occurred mainly near the inlet of the upper plate hole. In addition, the MC region was much larger than that of the other failure modes. Therefore, MC became the main failure mode during bolt insertion.

Photographs of the CFRP damage and the damage distribution obtained by simulation are shown in Figure 10. Both the FEA and tests showed that the same type of damage (MC) was observed at approximately the same location, with the entrance to the hole in the upper plate being the main region where damage occurred.

In this study, cohesive elements were used to simulate the interfacial debonding of CFRP HBB joints. Once the SDEG of a cohesive element reached 1 at all its material points, this element was removed (deleted). The removal of the cohesive element indicated the debonding of the interface between the two CFRP plates. In other words, the adhesive layer was gradually failing. Figure 11 showed the peak SDEG corresponding to 0.4%, 0.6%, 0.8%, and 1% interference-fit sizes, as well as the SDEG distribution around the joint hole at these interference sizes. In the case of the 0.4% interference-fit size, only a very small number of cohesive elements around the joint hole showed slight damage, with a maximum value of 0.3247 for SDEG. The increase in interference-fit size exacerbated the damage to the adhesive layer around the joint hole. The maximum values of SDEG were 0.7103 and 0.8585 at interference-fit sizes of 0.6% and 0.8%, respectively. Regarding the 1% interference-fit size, more damaged cohesive elements were observed around the joint hole and the maximum value of SDEG became 0.9096. As a result, the larger the interference-fit size, the more damaged cohesive elements around the joint hole and the higher the SDEG peak. However, for all interference-fit sizes in this study, the maximum value of SDEG did not reach the critical value of 1. In other words, the adhesive layer around the joint hole was damaged, but not completely, and still had bearing capacity.

## 5. Conclusions

This paper focused on the damage and failure mechanisms of CFRP HBB joints with different interference-fit sizes. Progressive damage analysis was performed in ABAQUS/Standard for CFRP HBB joints, based on which insertion force–displacement curves were obtained and the damage to CFRP plates and adhesive layers was investigated. In order to verify the correctness of the finite element model, a corresponding interference-fit bolt insertion test was also carried out. The simulation results were basically consistent with the experimental results. Some conclusions of this study are summarized as follows:The insertion force increased with increasing interference-fit size during bolt insertion. As the bolt insertion depth increased, the insertion force corresponding to a certain interference-fit size continued to increase. This force dropped a little when the bolt reached the adhesive layer between the two plates. The insertion force reached its maximum when the bolt completely penetrated the lower plate. Thereafter, this force decreased slightly and remained stable.For CFRP HBB joints, there was a close relationship between the interference-fit size and the damage to the composite laminate. The increase in interference-fit size caused an increase in the number of failure modes and led to an expansion of the failure region. These failures were caused by bolt pressure and interfacial friction and occurred mainly at the entrance to the joint hole of the upper plate. Since the fiber strength was much higher than the matrix strength, no fiber failure was observed at the four interference-fit sizes chosen in this study. Among the five failure modes that occurred, matrix compressive failure was the main failure mode during bolt insertion.Due to the increased size of the interference fit, the damaged region around the joint hole gradually expanded. The maximum values of SDEG were 0.3247, 0.7103, 0.8585, and 0.9096 at interference-fit sizes of 0.4%, 0.6%, 0.8%, and 1%, respectively. They were all below the critical value of 1, which indicates that the adhesive layer was damaged, but not completely, and still had loading capacity.

## Figures and Tables

**Figure 1 materials-16-03753-f001:**
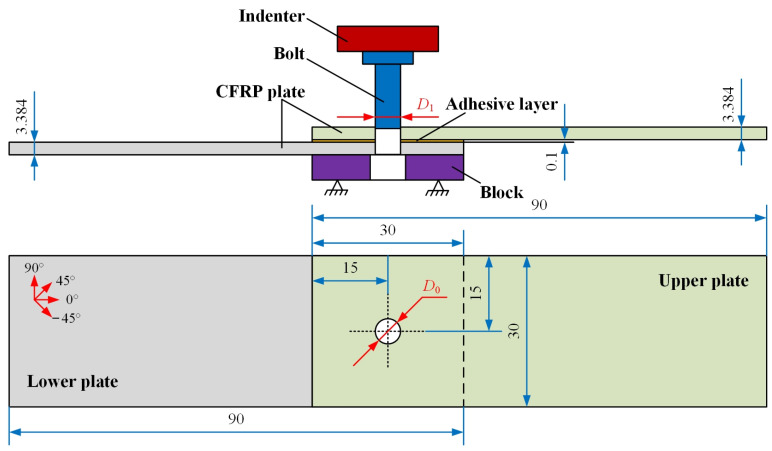
Schematic diagram of insertion specimen (unit: mm).

**Figure 2 materials-16-03753-f002:**
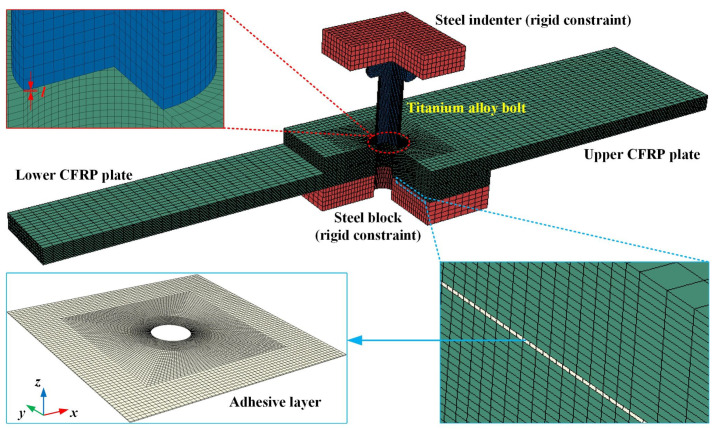
Finite element model of insertion specimen.

**Figure 3 materials-16-03753-f003:**
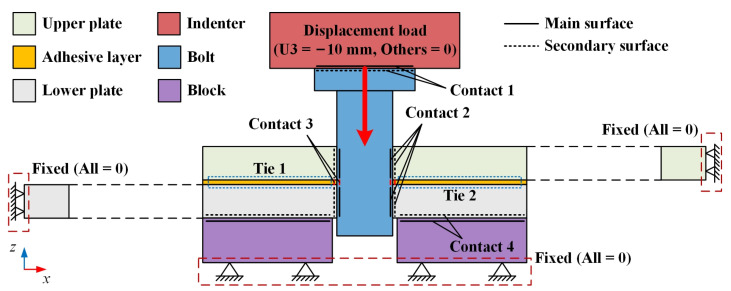
Contact relationships and boundary conditions.

**Figure 4 materials-16-03753-f004:**
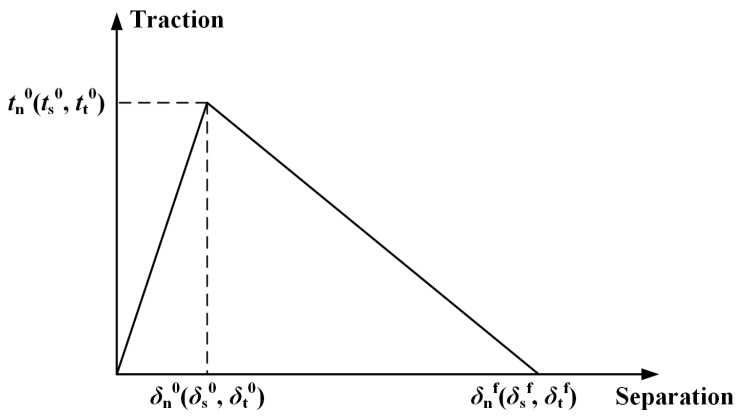
Bilinear constitutive model.

**Figure 5 materials-16-03753-f005:**
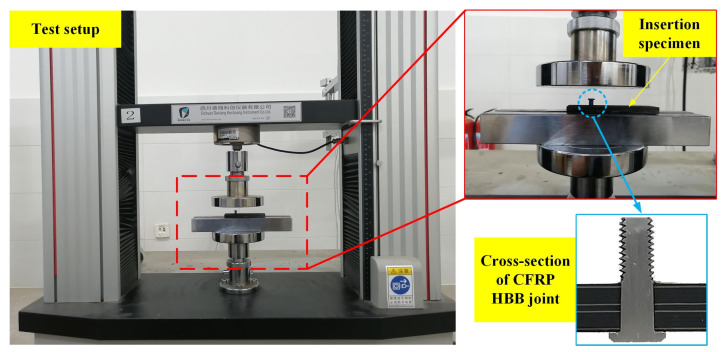
Test setup and insertion specimen.

**Figure 6 materials-16-03753-f006:**
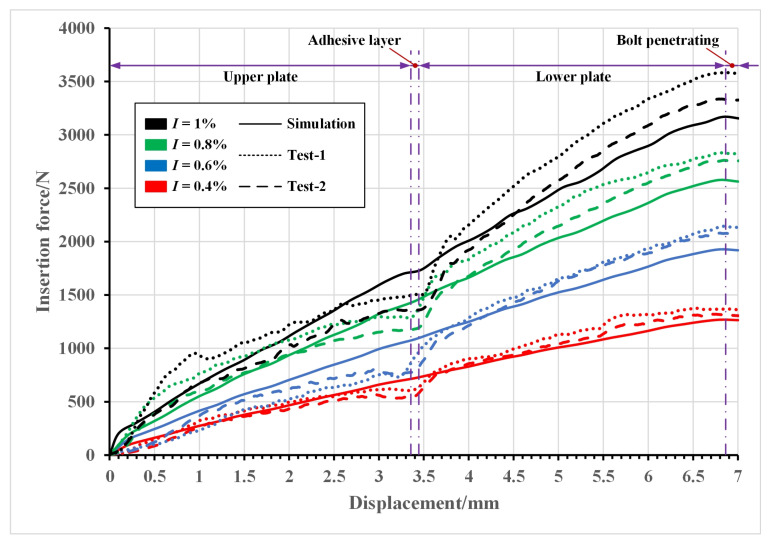
Insertion force versus bolt displacement under varying interference-fit sizes.

**Figure 7 materials-16-03753-f007:**
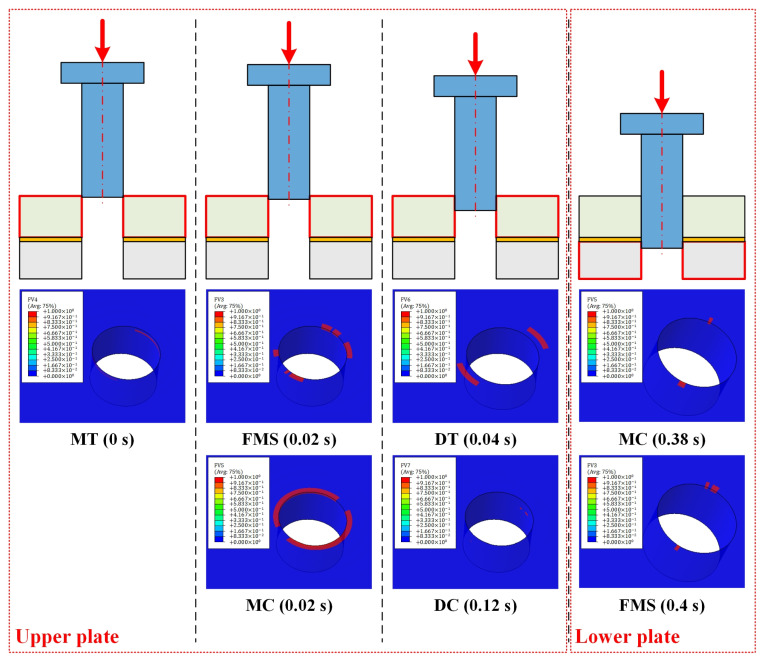
Failure sequence of the CFRP composite around the joint hole.

**Figure 8 materials-16-03753-f008:**
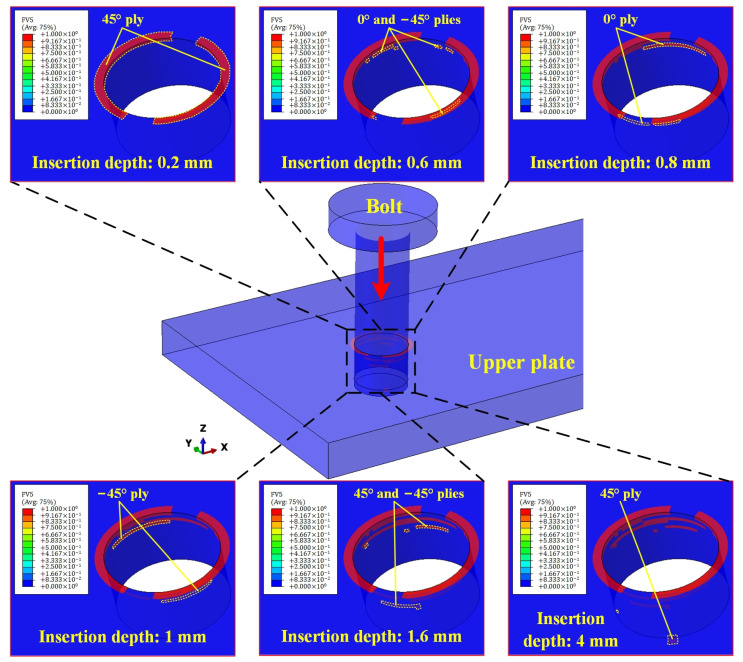
Expansion process of the failure region of the upper plate.

**Figure 9 materials-16-03753-f009:**
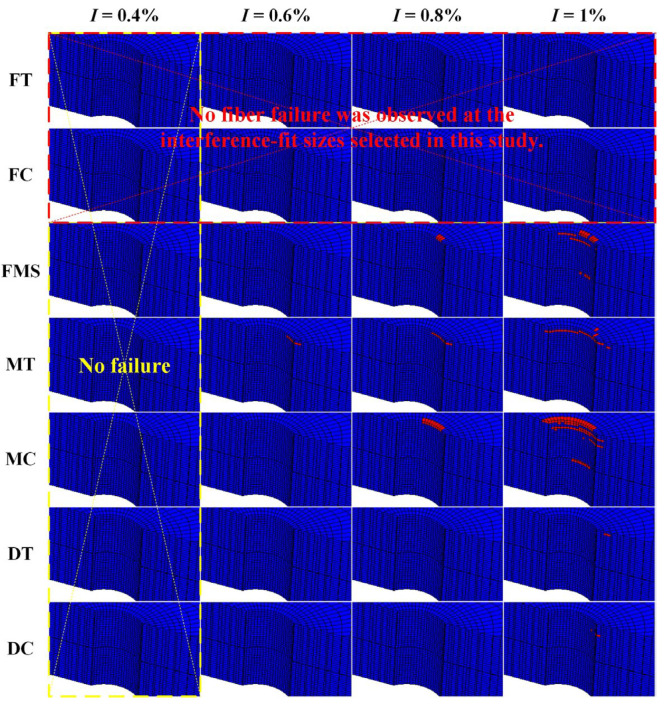
Failure region distribution of the seven failure modes around the hole for different interference-fit sizes.

**Figure 10 materials-16-03753-f010:**
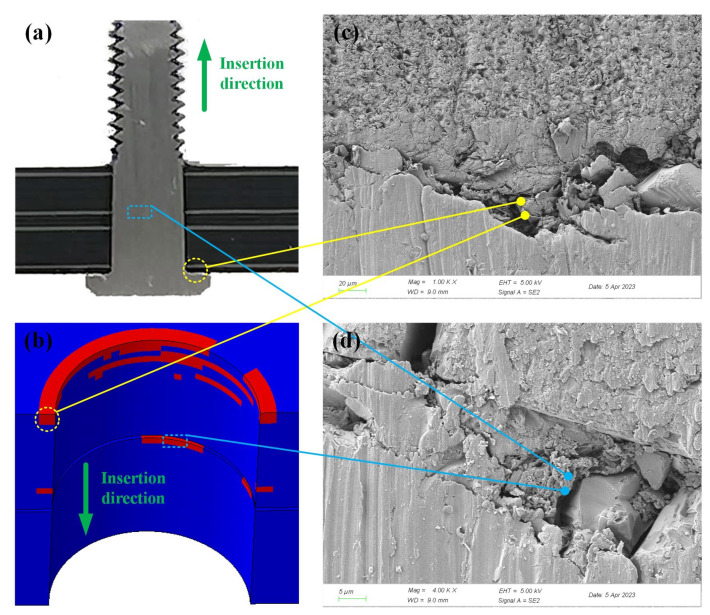
CFRP damage after bolt insertion (*I* = 1%): (**a**) CFRP HBB joint cross-section; (**b**) simulation results of MC; (**c**) damage at the inlet of the upper plate; and (**d**) damage observed in the lower plate.

**Figure 11 materials-16-03753-f011:**
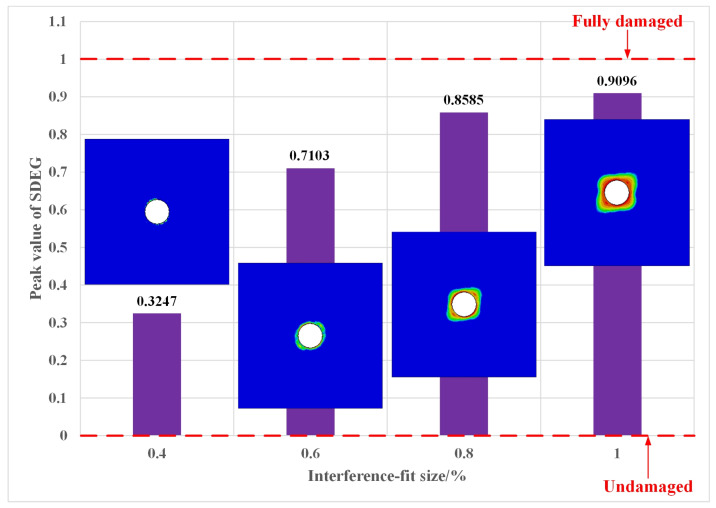
SDEG peak variation and SDEG distribution for different interference-fit sizes.

**Table 1 materials-16-03753-t001:** Material properties of CFRP laminates [31].

Elastic Modulus (MPa)			Poisson’s Ratio			Shear Modulus (MPa)		
*E* _11_	*E* _22_	*E* _33_	*v* _12_	*v* _13_	*v* _23_	*G* _12_	*G* _13_	*G* _23_
172,000	7000	7000	0.35	0.35	0.35	3900	3900	3900
**Tensile strength (MPa)**			**Compressive strength (MPa)**			**Shear strength (MPa)**		
*X* _T_	*Y* _T_	*Z* _T_	*X* _C_	*Y* _C_	*Z* _C_	*S* _12_	*S* _13_	*S* _23_
2630	62	62	1480	213	213	109	109	86

**Table 2 materials-16-03753-t002:** Mechanical properties of Hysol EA9361 [32].

Property	Value
Tensile modulus, *E*_nn_ (MPa)	5140
Shear modulus, *E*_ss_ = *E*_tt_ (MPa)	1740
Tensile cohesive strength, *t*_n_^0^ (MPa)	14.6
Shear cohesive strength, *t*_s_^0^ = *t*_t_^0^ (MPa)	27.5
Toughness in tension, *G*_n_^C^ (N∙mm^−1^)	1.0
Toughness in shear, *G*_s_^C^ = *G*_t_^C^ (N∙mm^−1^)	1.0

**Table 3 materials-16-03753-t003:** Property degradation rules.

Failure Mode	*E* _11_	*E* _22_	*E* _33_	*v* _12_	*v* _13_	*v* _23_	*G* _12_	*G* _13_	*G* _23_
No failure	1	1	1	1	1	1	1	1	1
Fiber tensile failure	0.07	1	1	1	1	1	1	1	1
Fiber compressive failure	0.14	1	1	1	1	1	1	1	1
Fiber–matrix shear damage	1	1	1	0	1	1	0	1	1
Matrix tensile failure	1	0.2	1	1	1	1	0.2	1	0.2
Matrix compressive failure	1	0.4	1	1	1	1	0.4	1	0.4
Delamination in tension	1	1	0	1	0	0	1	0	0
Delamination in compression	1	1	0	1	0	0	1	0	0

**Table 4 materials-16-03753-t004:** Geometric and material data of the specimen.

CFRP plate (T800/epoxy)	Stacking sequence: [45/−45/0/45/90/−45/45/90/−45]_s_ Single layer thickness: 0.188 mm Dimensions: 90 mm × 30 mm × 3.384 mm
Adhesive layer (Hysol EA9361)	Dimensions: 30 mm × 30 mm × 0.1 mm
Hi-lock bolt (Titanium alloy)	Diameter: 5 mm Length of bolt shank: 7 mm

**Table 5 materials-16-03753-t005:** Interference-fit data in this study.

Bolt Diameter (mm)	Nominal Value		Real Interference-Fit Interval (%)	
	Interference-Fit Size (%)	Hole Diameter (mm)	Min	Max
5.000 ± 0.010	0.4	4.980	0.25	0.57
	0.6	4.970	0.34	0.68
	0.8	4.960	0.56	0.87
	1	4.950	0.77	1.06

**Table 6 materials-16-03753-t006:** Test and simulation results of peak insertion force.

Interference-Fit Size (%)	Test Result (N)	Simulation Result (N)	Error (%)
0.4	1348	1269	5.86
0.6	2112	1928	8.73
0.8	2797	2578	7.83
1	3461	3169	8.45

## Data Availability

Not applicable.

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
