# Peer review of "Damage Analysis of CFRP Hybrid Bonded-Bolted Joint during Insertion of Interference-Fit Bolt"

_materials, 2023, doi:10.3390/ma16103753_

Round 1

Reviewer 1 Report

This paper presents an experimental and numerical analysis of damage a around the Joint-hole. Overall,  the submitted manuscript is well written and structured. I have no doubt that this work is relevant to a broader audience. I believe that this paper will be suitable for publication if the following recommendations are implemented.

- Authors are invited to write more explanations about the failure mechanism around the hole. Please refer to these papers (10.1177/0021998319857463, doi.org/10.1016/j.engfracmech.2021.107802).

- Please add the source of the used damage parameters in the numerical part (Table 1 and Table 2). if they are taken from previous work. If an identification procedure was applied, it should be part of the manuscript.

Reviewer 2 Report

The paper proposes a finite element simulation method for interference fit composite bolted joints. The manuscript is well-written and rigorously presented. Some aspects regarding the modeling approach and the validation of the results need to be improved. Therefore, the revised version of the manuscript should consider the following amendments:

·         Section 2.2 and Section 2.3 have the same title.

·         Section 2.3 – Considering the interference-fit behavior of the joint, why did you use a hard contact method? Adding the local surface deformation could be useful to obtain interesting results.

·         Section 2.4 – The rationale for the time variation of the interference is not clear. Why the interference provided with the contact module cannot remain constant during the simulation? How can be guaranteed that a varying value is meaningful and in agreement with the real behavior of the joint?

·         Section 3.1 – Avoid the repetition of the geometrical and material data outlined in Section 2. A Table with this information should be added and recalled.

·         Section 3.1 – Report the percentage interference and diameters value in a Table to make easier the reading.

·         Section 4.1 – Some contours of displacement taken from the finite element analysis should be shown to present the numerical results.

·         Section 4.2 – To increase the significance of the work, the results in terms of damage detection should be compared with the findings of the experimental tests. Did the tested specimens undergo the same kind of damage on the same layers and in analogous positions?

·         The literature review regarding the simulations of composite bolted joints need to be enlarged including more references  as: https://doi.org/10.1016/j.compstruct.2020.112005; https://doi.org/10.1016/j.prostr.2020.02.078; https://doi.org/10.1016/j.compstruct.2020.112770

The quality of English is good, only minor editing of the language is necessary.

Reviewer 3 Report

The submitted paper, "Damage analysis of CFRP hybrid bonded-bolted joint during insertion of interference-fit bolt," is informative. This paper presents the effect of interference-fit sizes on the damage of CFRP hybrid bonded bolted (HBB) joints during bolt insertion. This topic has not been widely researched in the field of composite materials. The research used experimental and numerical analysis (finite element analysis). This paper can be accepted for the Material Journal after adding a few points according to the following suggestions:

1. The paper does not explain the main question addressed by the authors in this study. It can be explained by giving examples of cases from this work in actual conditions.

2. There is no list of the specimens tested in this study in the form of a table. Add a table showing the test variables in the test.

3. The text does not show photos of the experimental test results. Please add to the text the photos showing the damage and failure of composite laminates.

4. Based on the test results and numerical analysis, it is not explicitly explained in conclusion the authors' suggestions for the test results of the CFRP hybrid bonded-bolted joint with different interference-fit sizes (0.4%, 0.6%, 0.8%, and 1%).

Reviewer 4 Report

This work investigated the effect of interference-fit sizes on the damage and failure of CFRP HBB joints. It is a good research study. The revision:

-        In order to provide a more comprehensive literature review, the authors should cite and discuss the following relevant papers in their revised manuscript:

Static capacity of tubular X-joints reinforced with fiber reinforced polymer subjected to compressive load. Engineering Structures. 2021 Jun 1;236:112041.

Nassiraei, H. and Rezadoost, P., 2021. SCFs in tubular X-joints retrofitted with FRP under out-of-plane bending moment. Marine Structures79, p.103010.

-        From Table 1, it can be seen that one CFRP type is used. Other type of CFRP (CFRP with other E1) are not investigated. How can the result be applied for the joints with other CFRP.

-        Please add mesh size on fig. 2. Also add sensitive analyses on mesh size.

-        How was the interface between the FRP layer and the steel members.

-        How was the FRP layer direction.

Round 2

Reviewer 2 Report

The required amendments were considered and implemented.

The paper can be accepted in present form.

Reviewer 4 Report

The authors have addressed all the comments to my satisfaction.